# Rehabilitation Machine for Bariatric Individuals

**Andrea Botta * , Paride Cavallone , Luca Carbonari, Carmen Visconte and Giuseppe Quaglia**

Department of Mechanical and Aerospace Engineering, Politecnico di Torino, 10129 Torino, Italy;
paride.cavallone@polito.it (P.C.); luca.carbonari@polito.it (L.C.); carmen.visconte@polito.it (C.V.);
giuseppe.quaglia@polito.it (G.Q.)
* Correspondence: andrea.botta@polito.it

**Abstract:** Obesity is known to be growing worldwide. The World Health Organization (WHO) reports that obesity has tripled since 1975. In 2016, 39% of adults over 18 years old were overweight, and 13% were obese. Obesity is mostly preventable by adopting lifestyle improvements, enhancing diet quality, and doing physical exercise. The workload of the physical exercises should be proportionate to the patient's capabilities. However, it must be considered that obese people are not used to training; they may not endure physical exertion and, even more critically, they could have some psychological impediments to the workouts. Physical exercises and equipment must, therefore, guarantee comfort and prevent situations in which the bariatric individual may feel inadequate. For these reasons, this study aims to design an innovative system to approach simple physical activities, like leg and arm exercises, to bariatric users to enable them to recover mobility and muscle tone gradually. The leading feature of this architecture is the design of hidden exercise mechanisms to overcome the psychological barriers of the users toward these kinds of machines. This paper proposes the initial design of the main sub-systems composing the rehabilitation machine, namely the leg curl and leg extension mechanism and its control architecture, the upper body exercises system, and a series of regulation mechanisms required to accommodate a wide range of users. The proposed functional design will then lead to the development of a prototype to validate the machine.

**Keywords:** rehabilitation machine; leg exercises; mechanism design; obesity and overweight rehabilitation

## 1. Introduction

Obesity is one of the world's most significant problems related to health, one that has shifted from being a problem in wealthy countries to one that affects all income levels. Globally, 39% of adults aged 18 years or more were overweight in 2016, and 13% were obese [1]. Between 2000 and 2018, in the US, the incidence of obesity has increased from 30.5% to 42.4%, and the prevalence of severe obesity has risen from 4.7% to 9.2% [2]. Obesity is a chronic and complex multifactorial disease caused by an excessive amount of body fat due to an imbalance between energy intake and consumption [1]. This pathology increases the risk of other diseases and health problems, such as cardiovascular diseases, diabetes, high blood pressure, and damage to the musculoskeletal system [3]. The positive news is that even moderate weight loss can improve or prevent obesity-related health issues. Behavioral changes, increased physical activity, and dietary changes lead to physical and psychological rehabilitation of obese individuals [1,4–6]. For more severe cases, prescription medications and weight-loss surgery procedures are used for treating the pathology [7,8].

The introduction of working out in obese individuals is not only about losing weight, but there are also other benefits, such as better mobility, improved self-esteem, and overall wellness. Several studies focused on which exercises are best suited for such individuals [9], agreeing that, in order to

increase the caloric expenditure, such exercises should involve the main and larger muscle groups. They should also be done in short sessions with minimal external load and repeated 5–7 days a week. Typical physical activity, but even the fidgeting or non-exercise activities, may lead to a relevant daily energy expenditure up to 350 kcal/day in obese individuals [10], reducing the risk of obesity and other health-related issues [11–14]. Koepp et al. [15,16] have observed an energetic expenditure increment ranging from 13 to 22 kcal/h while using commercial chair-fidgeting devices compared to sitting still. Even better results (comparable to a 2 mph walk) have been monitored when the user is engaged with exercise videos while fidgeting. Although these findings are promising, it seems that a full-body engagement may be required to obtain a significative increase in one's energy expenditure [17].

Particular care has to be paid to correctly perform the workout to avoid overloading and damaging the already weakened musculoskeletal system. For this reason, seated exercises, or more general exercises where it is possible to avoid loading the individuals' articulations with their whole weight, must be preferred. While these issues can be easily overcome with proper workout planning made by professionals to avoid physical stress, the crucial issue related to performing the exercises is the psychological stress associated with them [18–21]. Due to the physical condition of bariatric individuals, the workout can appear unsustainable, painful, and stressing, leading to stepping away, partially or totally, from the needed exercise routine. Moreover, obese individuals may feel a great deal of social pressure and prejudice if they have to perform their exercises where there are other people, leading again to distancing themselves from physical activities further. Secondly, the typical exercise machines can be felt like a cage or not sturdy enough to stand the weight of obese individuals.

Since it is essential to achieve an active routine that has to be carried out for a long time, this paper proposes a novel rehabilitation machine for bariatric individuals to introduce such people to light and simple physical exercises. This machine aims to enable them to recover muscle tone and mobility gradually and to softly approach this new healthy exercise routine. The key requirement leading the design of this machine is the will to overcome the psychological barrier of obese individuals toward such machines, to hide all the moving parts and mechanisms, and to provide some positive and motivational feedback. In short, this design aims at developing a rehabilitation machine that does not look like typical gym equipment, but it appears as a chair or an armchair.

The design of the rehabilitation machine has started from the guidelines and requirements provided by doctors of the medical science department of Università degli Studi di Torino. Thanks to their involvement, it has been possible to identify the proper exercises that can be done in a seated position, and that can be helpful to the rehabilitation process. Two antagonist leg movement exercises (leg curl and leg extension) have been identified as main exercises of the machine in order to involve a large muscle group (sartorius, gracilis, gastrocnemius, and femoral quadriceps). The upper body muscle groups (chest, shoulders, and arms) and the abdominal muscles are the secondary and tertiary involved muscles that can do some workout. Given these requirements, the design of the machine started with the synthesis of the mechanisms that enable these movements and some auxiliary regulations, followed by a preliminary study on a possible control system for the leg exercises and an initial executive design of the whole machine.

The main contribution of the paper is to present a novel solution conceived considering both the user point of view and the mechatronics requirements. The presented design can also be considered an example of a service robot, i.e., an automatic and controlled mechatronics machine that provides physical service to a human being. In this case, many variables (trajectories, forces) can be controlled according to the different exercise types, many parameters can be adjusted considering the user needs, and a human-machine interface can be defined to obtain the final results, i.e., improving the user condition. Moreover, the proposed device can provide many monitoring data to the therapist to evaluate the activities' effectiveness. Future developments also include an improved user interface able to implement some sort of "gamification" in order to motivate the user to exercise even more.

In the following section, the key steps of the design process of the main sub-systems are reported. After that, the results section collects and discusses the results of the leg exercises mechanism and its

actuation and control system since it is the most interesting sub-system in the machine. The conclusions section closes the paper, collecting and summarizing the discussion on the findings and results.

## 2. Rehabilitation Machine Design

The design of exercise or rehabilitation machines is not a novelty; indeed, there are probably thousands of machines optimized for these tasks. The optimization processes behind these machines led to very efficient mechanisms able to perform the desired exercises correctly. Nevertheless, little, if not at all, care has been paid to possible physical or psychological barriers of their users. A typical user of such pieces of equipment is already motivated to use such machines. Therefore, the psychological aspect is a minor concern of the designer of these machines. However, as introduced before, several studies have reported that obese individuals may avoid using conventional exercise machines due to physical limitations, but mostly due to psychological ones. Similar claims have come from doctors working with bariatric patients at the medical science department of Università degli studi di Torino.

The idea behind this study is to design a rehabilitation machine to enable obese individuals to recover muscle tone and mobility gradually and to softly approach this new healthy exercise routine. However, in trying to achieve this objective, most of the design choices are guided by the concepts of overcoming all the possible psychological barriers of bariatric users and of motivating them to continue their rehabilitation routine. As stated before, the appearance of the machine is key to achieve this aim. However, it is important to state that this work does not want to propose aesthetically appealing machines since this departs from the research team competences. Instead, the actual objective is to design novel mechanisms that enable the user to correctly perform some physical exercises but with the requirement that such mechanisms can be hidden while the machine is in the rest position.

The following subsections address all the requirements and guidelines of the whole machine and its most relevant sub-systems. After that, the synthesis and design of the main mechanisms of the rehabilitation machine are presented. A proposal of the actuation for the leg mechanism control system is discussed at the end of this section.

### 2.1. System Requirements

The idea is to design a rehabilitation machine that appears like an ordinary object, like a chair. This concept should ease the introduction of obese individuals to physical exercises. Figure 1 illustrates the concept of the prototype discussed here. At first sight, the machine appears as a chair for bariatric people (i.e., a chair larger than the common chairs in order to accommodate obese individuals). However, under the seat, it is possible to see the mechanisms that enable the physical exercises.

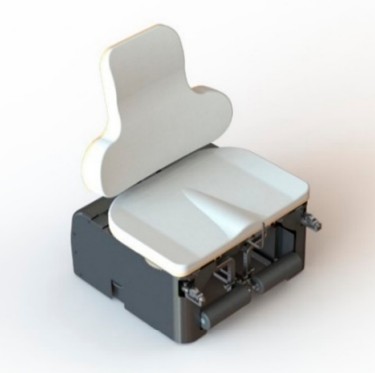

**Figure 1.** Concept of the prototype of the rehabilitation machine.

The second requirement coming from the medical department is which exercises an obese individual should do. Following the findings of [9,22,23], the most suitable exercises for rehabilitation of bariatric people have to involve the main and larger muscle groups in order to increase the caloric

expenditure. Such exercises should be done in short sessions repeated regularly during the week, almost every day. The load of the exercises should be minimal since aerobic exercises should be preferred and also because the already weakened musculoskeletal structure of obese individuals must not be overloaded and damaged. Obese individuals performing the exercises should avoid loading their articulation with their weight (e.g., doing squats) in order to prevent any damage to their joints. For this reason, seated exercises, or more general exercises where the load applied is only due to an external and known source, should be preferred.

Given these guidelines, the doctors identified the proper exercises involving legs, abdomen, and the upper torso (chest, shoulder, arms), some of the largest muscle groups. More details about the actual exercises will be presented in the following subsections. Whatever the exercises, the machine must permit its user to perform the movement correctly to avoid any damage to muscles or joints. The exercise loads must be controllable by the user, following a protocol defined by the physiotherapists, and it has to be constant during the exercise. The user-exerted force and other useful quantities, such as the movements velocity and energy consumption, must be measured and logged. This feature could give doctors some insight or could motivate its user, for example, through some sort of gamification of the whole exercises.

The last general requirements are linked to the size of the machine. The machine has to stand the weight of bariatric people; therefore, the system must stand a weight of about 400 kg. It is also central that the machine can accommodate users of different sizes. For this reason, the machine should have several auxiliary regulations in order to adapt the machine to a wide range of individuals. The actual references for such dimensions come from [24], and actual measurements performed on 20 bariatric patients at Molinette hospital in Turin. The final requirement about the overall size of the machine is that the machine must be able to go through hospital doors (approx. 80 cm) and the main doors in the home (approx. 75 cm).

### 2.2. Leg Exercises Mechanism

The main exercises that can be done with this machine involve the legs, since some of the largest group muscles are in the lower body. In particular, the exercises are seated leg curls (the flexion movement of the legs) and seated leg extensions, two antagonist exercises involving sartorius, gracilis, gastrocnemius, and femoral quadriceps muscles. Figure 2 depicts a schematic representation of the two exercises. In the figure, but also in designing the machine, an ideal rotational joint, named $K^*$, approximates the knee joint. Such hypothesis is present in all common gym equipment, even if the actual knee joint is more complicated than that, and the instantaneous center of rotation (ICR) of the leg is not a fixed point.

To guide the leg correctly during the exercises, the ICRs of the leg and of the mechanism links moving with it must be the point $K^*$. In typical machines, this is done by placing a rotational joint in $K^*$. By doing so, however, the mechanism would stick out of the chair, being a possible obstacle to the user. The mechanism proposed here instead has a virtual joint at the ICR $K^*$, but the whole mechanism lies under the seat when entirely retracted in the rest position. Figure 3 shows the functional scheme of the mechanism composed by a double parallelogram. In order to have the mechanism ICR in $K^*$, the revolute joints $B_0$ and $A_0$ fixed to the seat and the point $K^*$ must be along of the same line. The same must be true also for the joint $C_1$, $C_2$, and the point $K^*$. With this configuration, the mechanism can follow the leg along its rotation about $K^*$. In order to accommodate a wide range of users, two prismatic joints are present in $P$ and $R$ to adjust the point $R$ of the mechanism where can be fixed the user-machine contact device, i.e., a cylindrical cushion. The figure shows the two alternative positions of the cushion that can be set in order to perform the two different exercises: leg curl (the cushion is behind the ankle) and leg extension (the cushion is in front of the ankle). A linear actuator, connected between the revolute joints $T_0$ (fixed to the frame) and $T$, is controlled to exert a constant and desired torque about $K^*$ that the user has to match applying a force in $R$ with their leg. It is important

to highlight that the ICR $K^*$ is fixed with respect to the structure and not with the user; therefore, some auxiliary regulations are required to align the knee with this point.

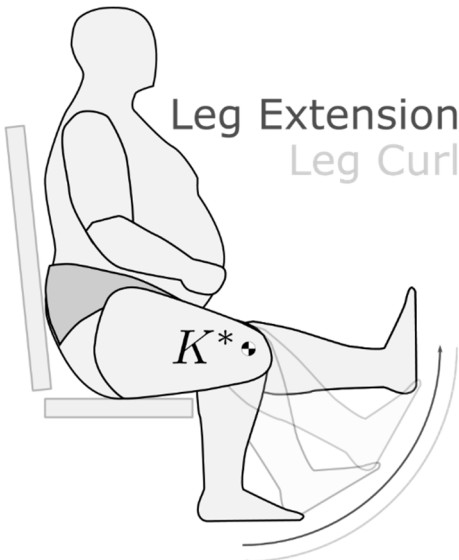

**Figure 2.** Representation of the seated leg curls and leg extensions.

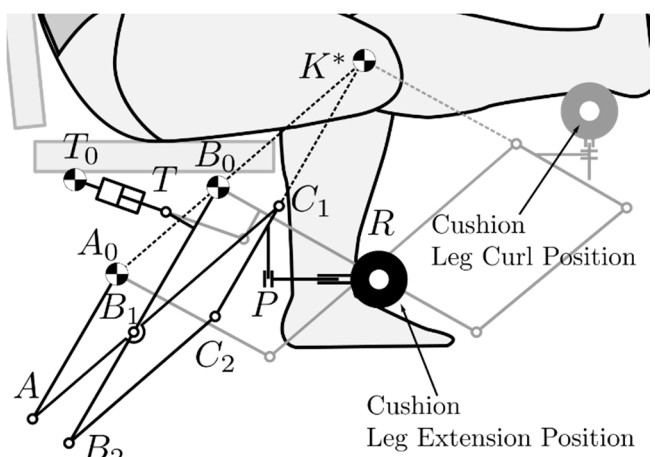

**Figure 3.** Leg exercise mechanism in its two extreme configurations. Alternative cushion positions are shown.

During both exercises, the leg has to span an angle $\Delta\theta = 90°$. Therefore, the link $C_1C_2$ rotates around the same angle, since it is rigidly connected to the user leg (when the regulations in $P$ and $R$ are fixed). Moreover, given the mechanism architecture, the $A_0A$ and $B_0B_1B_2$ links must rotate around $\Delta\theta$ too because they are always parallel to $C_1C_2$. It is supposed that the maximum force produced by the mechanism in $R$ is about 250 $N$ for each leg, with an average angular velocity of the leg of 1.5 rad/s (one repetition per second). The opposing force must be constant during the movement; therefore, the instantaneous transmission ratio of the mechanism has to be as constant as possible. The last requirement is that the mechanism must be contained under the seat when it is completely retracted.

Figure 4 shows the mechanism with its main parameters and its variables. Table 1 collects symbols and descriptions of such quantities. Figure 5 represents the optimal configuration of the actuator in the mechanism: in order to obtain an instantaneous transmission ratio as constant as possible, the points $T_0$, $T_{initial}$ and $T_{final}$ must be aligned. The other related parameters are then optimized in order to improve the mechanism performance but also to minimize the dimensions of the whole

mechanism. Given this architecture and a movement that spans 90°, the linear actuator stroke $\Delta l_{act}$ is always $\sqrt{2}\, r$.

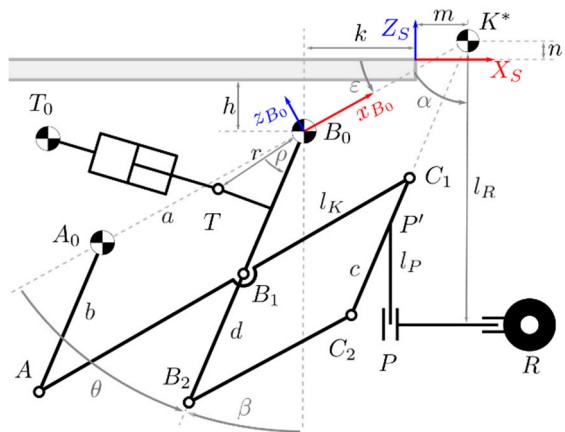

**Figure 4.** Parameters and variables of the mechanism.

**Table 1.** Parameters of the leg exercises mechanism.

| Symbol | Description | Symbol | Description |
|---|---|---|---|
| $a$ | Length of $\overline{A_0 B_0}$ | $m$ | Abscissa of $K^*$ in $S$ |
| $b$ | Length of $\overline{A_0 A}$ and $\overline{B_0 B_1}$ | $n$ | Ordinate of $K^*$ in $S$ |
| $c$ | Length of $\overline{P'C_2}$ | $r$ | $TB_0$ distance |
| $d$ | Length of $\overline{B_1 B_2}$ and $\overline{C_1 C_2}$ | $\theta_0$ | Angle between $x_{B_0}$ and link $B_0 B_1 B_2$ when the machine is retracted |
| $-h$ | Ordinate of $B_0$ in $S$ | $\theta$ | Angle of the exercises |
| $-k$ | Abscissa of $B_0$ in $S$ | $\nu$ | Angle between the $T_0 T$ direction and $X_S$ |
| $l_{act}$ | Length of the actuator, length of $\overline{T_0 T}$. $l_{att,0}$ is the initial value | $\rho$ | Angle between $x_{B_0}$ and $TB_0$ direction |
| $l_R$ | $K^*R$ distance | | |

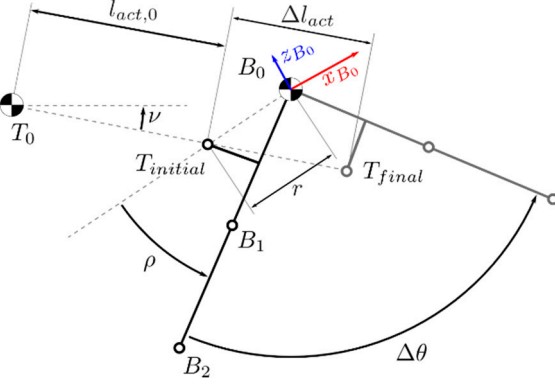

**Figure 5.** Detail about the actuator configuration.

From Figure 4 it is also possible to derive the angle $\varepsilon$ between $X_S$ and $x_{B_0}$, the angle $\beta$ between $Z_S$ and the link $B_0 B_1 B_2$, the angle $\nu$, and the last unknown length of the mechanism $l_k$:

$$\varepsilon = \tan^{-1}\left(\frac{n+h}{m+k}\right) \tag{1}$$

$$\beta = \frac{\pi}{2} - \epsilon - \theta \tag{2}$$

$$v = \tan^{-1}\left(\frac{Z_T - Z_{T_0}}{X_T - X_{T_0}}\right), v_0 = v_{final} = \tan^{-1}\left(\frac{Z_{T,final} - Z_{T,initial}}{X_{T,final} - X_{T,initial}}\right) \tag{3}$$

$$l_k = \overline{B_0 K^*} = \overline{B_1 C_1} = \overline{B_2 C_2} = \sqrt{(m+k)^2 + (n+h)^2} \tag{4}$$

Given these new quantities, it is possible to obtain the relative positions of each point of the mechanism. However, since such relations are easily obtained using trigonometry, they are not shown here. The most relevant relation that can be obtained from the cited ones is the length $\overline{T_0 T}$ as a function of $\theta$, or, in other words, the relation between the rotation of the system $\theta$ and the length of the actuator $l_{act}$

$$l_{act} = \sqrt{(l_{act,0} \sin v + r(\sin(\sigma + \theta_0) - \sin(\sigma + \theta)))^2 + (l_{act,0} \cos v + r(\cos(\sigma + \theta_0) - \cos(\sigma + \theta)))^2} \tag{5}$$

where $l_{act,0} = l_0 + r\sqrt{2}$ and $\sigma = \varepsilon - \rho$. $l_0$ is a fixed length of the linear actuator that depends on the specific commercial model. The chosen actuator needs to satisfy the initial length $l_{act,0}$ and the maximum length $l_{act,max} = l_{act,0} + \Delta l_{act}$.

The derivative of Equation (5) about $\theta$ is the mechanism transmission ratio:

$$\frac{\delta l_{act}}{\delta \theta} = \frac{r(l_{act,0}\sin(\sigma - v + \theta) - r\sin(\theta_0 - \theta))}{l_{act}} \tag{6}$$

By knowing these relations and the requirements listed before, it has been possible to perform an optimization of the mechanism parameters by an iterative procedure in order to satisfy the requirements while obtaining a feasible solution mostly composed of typical commercial parts.

From the diagram in Figure 6, it is already possible to derive a first approximation of the actuator force $F_{act}$ that it is required to balance the user force $F_u$. The figure shows the leg extension case, but the leg curl exercise is analogous but with forces with opposite signs. For this initial approximation, just the actuator and user forces are considered. Here it is made the hypothesis that the user applies a constant force $F_u$ in $R$ that it is perpendicular to the link $P'P$. Therefore, it is like a torque equal to $F_u l_R$ is acting on the mechanism. In order to oppose the user movement, the machine must generate the force $F_{act}$. The force balance is the following:

$$F_u l_R = F_{act} r \sin(\varepsilon + \theta - \rho + v) = F_{act} r \sin(\gamma + v) \tag{7}$$

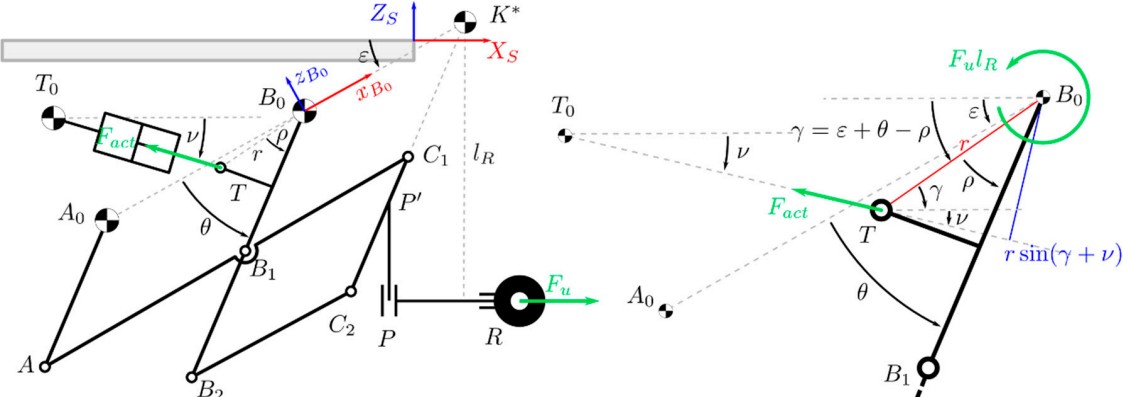

**Figure 6.** Machine dynamics for the leg extension exercise. For the leg curl exercises, the forces have opposite directions. On the right, there is the detail about the angles.

From this simple case, it is then possible to size the mechanism links, joints, and the actuator. After that, it is possible to refine the initial approximation taking into consideration the links weight

and the effect of an elastic element required to compensate the weight of the mechanism at rest. This more precise model can be expressed as:

$$F_u l_R + T_{weight} - T_{elastic} = F_{act,corr} r \sin(\gamma + \nu) \tag{8}$$

where $T_{weight} = T_{weight}(\theta)$ is the equivalent torque applied to the link $B_0 B_1 B_2$ generated by the effect of the weight of the whole mechanism, $T_{elastic} = T_{elastic}(\theta)$ is the torque applied by an elastic element to the same link in order to balance the weight when the mechanism is closed, $F_{act,corr}$ is the actuator force required to balance all the other components. $T_{weight} = T_{weight}(\theta)$ is easily derived, after the first definition of the links sizes and weights, computing the equivalent torque due to each mass with respect to the pivot point $B_0$. By knowing the value of $T_{weight}$ at $\theta_0$, it is possible to design the elastic element characteristics (stiffness, preload, free-length) in order to balance the weight when no external force is applied. One end of the spring is fixed to the chair structure, while the other is fixed to the link $B_0 B_1 B_2$.

### 2.3. Leg Exercises Control System

A pneumatic cylinder is used to apply the force required to oppose the user effort. Pneumatic actuation systems are commonly used in several gym equipments because it is possible to achieve the desired force with consistency and precision, regulating the pressure of the system. Moreover, such systems have minimal weight inertial effects during the exercise movement compared to weight-based gym machines for the same exercises.

Figure 7 shows the working principle of the pneumatic system used in this mechanism. During the exercise, only one chamber of the cylinder is under pressure, while the other one can be directly connected to the environment. It is possible to select the desired exercise, leg curl or leg extension, by switching the active chamber. The active chamber is connected to an accumulator through a normally closed proportional flow valve. The cylinder moving during the exercise reduces the volume of the active chamber, increasing the pressure within and, therefore, the generated resisting force. It is possible to control the cylinder pressure $p_{cyl}$ regulating the airflow from the chamber to the accumulator (that it is always kept at lower pressure) acting on the proportional flow valve. If the accumulator volume is much larger than the active chamber one, ideally infinitely large, it is possible to minimize the times when a compressor has to be used to keep the pressure to the desired set value, because its pressure stays almost constant.

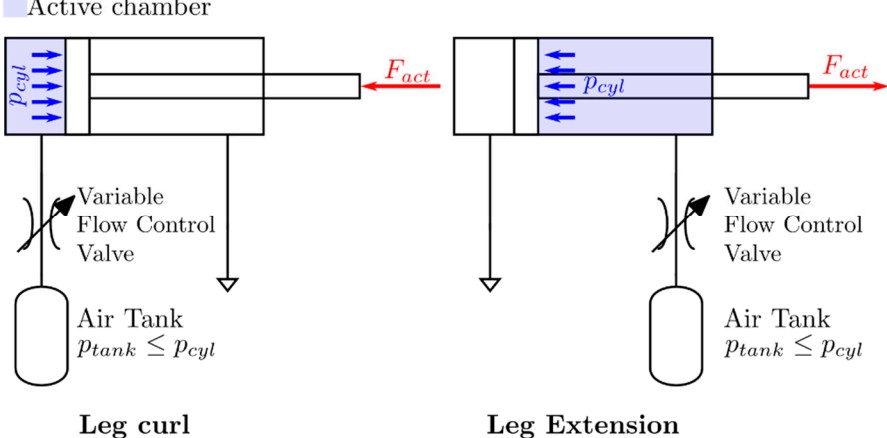

**Figure 7.** Pneumatic actuation system working principle.

This layout has been chosen in order to avoid high energy consumption and noise issues due to a full pressure control in the actuator chambers using, for example, controllable proportional valves.

In this case, the initial pressure of the tank $p_{tank,0}$ is set depending on the desired force $F_u$, then during the exercises the proportional flow valve is modulated to obtain the required $F_{act}$.

From Equation (8), it is possible to obtain $p_{set}$, the set pressure of the cylinder chamber required to maintain a constant resistance to the user movement:

$$F_u l_R + T_{weight} - T_{elastic} = p_{set} A_{eff} \, r \sin(\gamma + \nu) \tag{9}$$

$$p_{set} = k_{corr} \frac{F_u l_R + T_{weight} - T_{elastic}}{A_{eff} \, r \sin(\gamma + \nu)} \tag{10}$$

where $k_{corr}$ is a corrective factor used to model the internal actuator losses. Due to the cylinder rod, the effective areas $A_{eff}$ of the two chambers are slightly different; therefore, the set pressures for leg curl or leg extension exercises with the same reference force are slightly different.

Figure 8 shows a possible pneumatic circuit. Starting from the concept presented before, the required valves, the sensors, a compressor, and a control unit are added to obtain a functional pneumatic circuit to actuate and control the left and right leg exercises mechanisms. A 2/2 proportional valve works as a proportional flow control valve, while 4/2 valves are used to select which exercise perform, leg curl or leg extension. Pressure sensors are used to keep track of the controlled pressure closing the loop of the control system and to monitor the pressure in the circuit to avoid failure. Angular sensors measure the mechanism angular displacement in order to compute all the elements of Equation (10) and, therefore, to define the reference pressure $p_{set}$. Silenced exhausts are implemented to minimize the machine noise. A compressor, a pressure regulator valve, and a exhaust valve guarantee that the pressure within the circuit is kept at a defined value slightly lower than the minimum value of $p_{set}$. This condition is required to maintain the correct flow direction from the cylinder to the air tank, because, in the real pneumatic circuit, the accumulator pressure increases during the exercises.

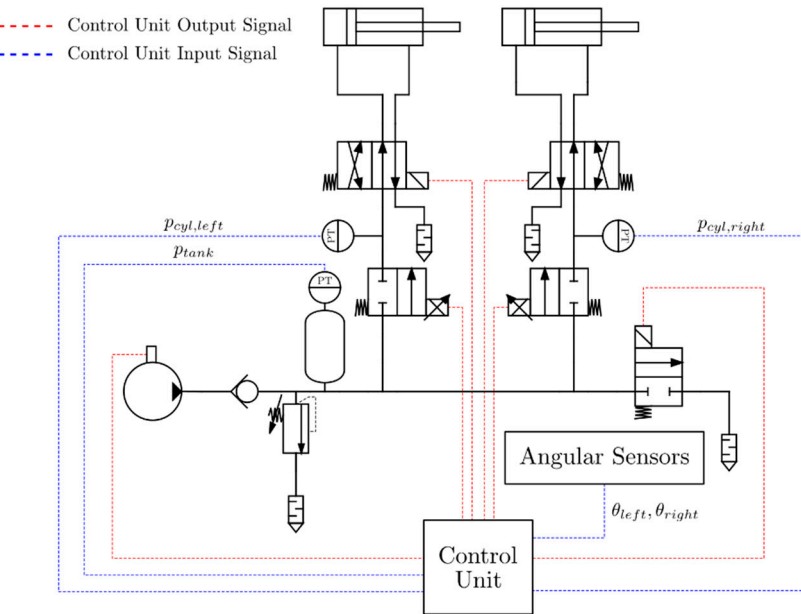

**Figure 8.** Pneumatic circuit of the leg exercises mechanisms.

In Section 4, some simulation results related to kinematic and dynamic performances of the system are reported.

### 2.4. Upper Body Exercises Mechanism

Figure 9 depicts the second system designed in order to exercise the upper body of bariatric individuals. A cable system enables the user to several exercises for the arms, shoulders, and chest.

This architecture is highly inspired by the innovative wheelchair propulsion system proposed in [25,26] because it allows its user greater mobility while pulling the cables.

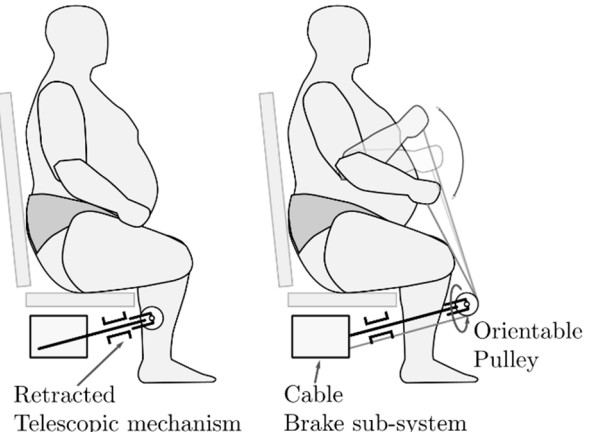

**Figure 9.** Upper body exercises system in its two configurations.

Like for the leg exercises mechanism, the user sets a desired opposing force to their exercises. An electronically controlled brake keeps this force constant as desired. In parallel to the brake, a torsional spring acts as a cable rewinder. Figure 9 shows a single orientable pulley in the cable system that is required to perform many exercises without difficulty. Depending on the space under the seat, other pulleys can be added to guide the cable to the braking system correctly.

The upper body exercises mechanism follows the same principle of the whole machine: the machine should not appear as gym equipment, but it should look like an ordinary chair. For this reason, a telescopic bar enables to hide the mechanism under the seat when not in use. When required, the telescopic bar can be extracted and locked to position to perform the workout.

The development of this sub-system is still at an early stage, and it is not fully defined. However, the concept is to develop a sub-system that can oppose a controlled and constant resistance to the cable pulling. Several actuator systems could be suitable to perform this task, but the one that could be more interesting is a magnetic particle brake system. Magnetic particle brakes are commonly used in cable tension control because they can apply a torque very accurately with a fast response.

*2.5. Structure and Regulation Mechanisms*

The final step of the rehabilitation machine is the design of its structure and of some regulation mechanisms to enable individuals to correctly carry out the exercises and to accommodate a wide range of different individuals.

The first requirement of the structure is obviously to support the weight of the user and the whole machine. However, the structural elements of the machine are mainly made of bars, rods, and profiles; therefore, most of the weight is due to the user and, given the pathology of the potential users of the machine, this weight can be considerably high. The maximum weight of 400 kg has been considered to size the structure with a proper safety factor. The weights distribution of the user and the machine leads to another requirement of the structure. The base of the structure must be wide enough to guarantee the stability and avoid rollovers also during the exercises when relevant masses are in motion. Nevertheless, at the same time, it is desirable to minimize the size of the base and at least guarantee that the machine can go through hospital and main household doors. The last requirement related to the size of the machine is about the free space under the seat. The free space under the seat should be large enough to integrate all the required sub-systems. This volume can be approximated by a rectangular box with a base of $650 \times 690$ mm and a height of 400 mm when completely retracted. Hence, the components that not require direct interaction with the mechanism or the structure are integrated into a separated volume that is connected with the machine through pneumatic lines and

electrical connections. Parts of these components can be the air tank, the compressor, the control unit, and the human-machine interface.

Individuals of different body sizes could use the machine; hence, it is mandatory to adapt the machine to the user by lifting and rotating the seat and by adjusting the back of the chair. These regulations are needed to allow the user to perform his exercise correctly and comfortably. Moreover, it is useful to underline again that the machine should look and work, as close as possible to a chair, in particular when it is not used to do physical activities.

A scissor-lift mechanism lifts the seat in order to accommodate the shortest and tallest individuals. From the anthropometric data, the increment in height should be at least 120 mm. The same mechanism also acts as a structure; hence it is the same on the left and right side of the machine. The two sides are linked together to improve stiffness and stability and to avoid any asynchronous motion of one side relative to the other.

Figure 10 display the schematic representation of the scissor-lift mechanism in its two extreme configurations. The revolute joint $D_0$ and the sliding revolute joint $E_1$ are fixed to the bottom structure, while the sliding revolute joint $D_1$ and the revolute joint $E$ are placed into the upper one, where the seat is fixed. Acting on a linear actuator fixed with two revolute joints in $E_1$ and $E_2$ it is possible to change the height of the seat. Several variations of the actuator placement had been considered, but, for the sake of simplicity, more empty volume under the seat, and slightly better performance, only this one is presented here. Table 2 collects the parameter useful to define this mechanism. In the figure, there are other angles useful to write the input-output relation compactly.

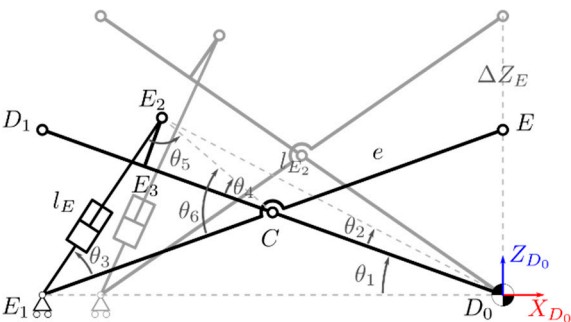

**Figure 10.** The scissor-lift mechanism in its two extreme configurations.

**Table 2.** Parameters and variables of the scissor-lift mechanism.

| Symbol | Description |
| --- | --- |
| $e$ | Half-length of $\overline{D_0D_1}$ and $\overline{E_1E}$ |
| $l_E$ | Variable length of the actuator, length of $\overline{E_1E_2}$ |
| $l_{E_2}$ | Length of $\overline{D_0E_2}$ |
| $-X_{E_1}$ | Abscissa of $E_1$ |
| $Z_E$ | Ordinate of $E$ and $D_1$. Height of the seat from the ground |
| $\theta_1$ | Variable angle between $X_{D_0}$ and link $D_0D_1$ |
| $\theta_2$ | Fixed angle between link $D_0D_1$ and the $D_0E_2$ direction |

From Figure 10, it is possible to obtain:

$$l_{E_3} = \overline{E_2E_3} = \sqrt{e^2 + l_{E_2}^2 - 2el_{E_2}\cos\theta_2} \qquad \theta_5 = \mathrm{acos}\left(\frac{e^2 - l_{E_3}^2 - l_E}{-2el_{E_3}}\right)$$

$$\theta_3 = \mathrm{asin}\left(\frac{l_{E_3}}{e}\sin\theta_5\right) \qquad \theta_6 = \pi - \theta_3 - \theta_5 \tag{11}$$

$$\theta_4 = \mathrm{atan}\left(\frac{l_{E_2}\sin\theta_2}{l_{E_2}\cos\theta_2 - e}\right) \qquad \theta_1 = \frac{\theta_6 - \theta_4}{2}$$

knowing $\theta_1$ as a function of the geometric parameters of the mechanism and of the variable actuator length, it is possible to derive the relation between the seat height and the actuator length:

$$Z_E = 2e \sin \theta_1 \tag{12}$$

From this equation, it is possible to optimize the parameters with an iterative process with the purpose of design a mechanism that can lift the seat of at least 120 mm with a feasible actuator stroke. The instantaneous transmission ratio that, in this case, is evaluated only numerically, and should be as constant as possible to obtain a smooth motion.

Given all the previous angles, it is also possible to evaluate the force required to stand a user with a weight of $P = 4000 \, N$ using the free body diagram of the mechanism derived from Figure 11.

$$F_E = \frac{2Pe \sqrt{\tan^2(\theta_1 + \theta_3) + 1}|\cos \theta_1|}{\left|2e \sin \theta_1 + l_{E_2}(\sin(\theta_1 + \theta_2) + \cos(\theta_1 + \theta_2) \tan(\theta_1 + \theta_3))\right|} \tag{13}$$

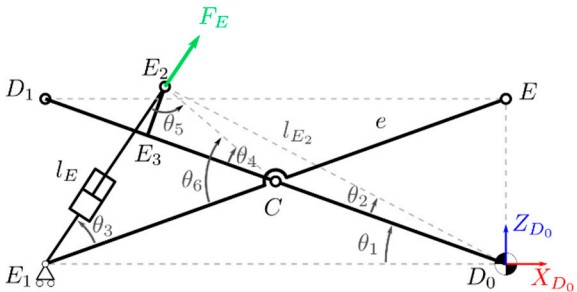

**Figure 11.** Force acting on the scissor lift mechanism.

It is clear from the equation that it is irrelevant where the load is applied from the actuator perspective.

The back of the seat implements two degrees of freedom as regulations. A translational motion sets the depth of the seat to align the knee of the user correctly to point $K^*$ (Figure 12). One end of a linear actuator, $T_{40}$, is fixed to the upper structure while the other one, $T_4$, is fixed to a structure where the back of the seat is mounted. The actuator moves a structure where the revolute joints $T_1$ and $T_3$ are fixed. The back of the chair rotates about $T_3$ employing another linear actuator fixed between $T_1$ and $T_2$ (Figure 13). This second degree of freedom allows the user to improve the chair comfort, but it also enables the possibility of doing some abdominal exercises if the back is reclined completely. The seat depth spans from 420 to 550 mm from the reference $S$, while the back can rotate up to $45°$ from the vertical direction.

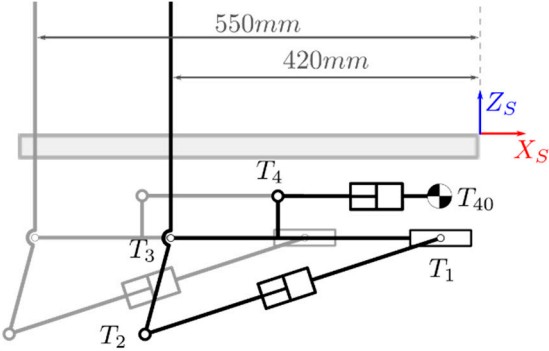

**Figure 12.** Seat depth regulation mechanism.

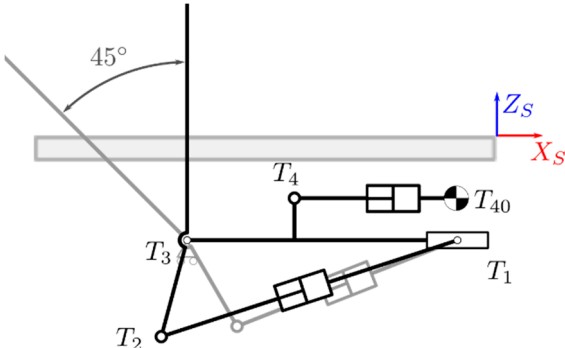

**Figure 13.** Back of the seat inclination mechanism.

The final regulation allows tuning the alignment between the knee of the user and the point $K^*$ finely. Point $K^*$ is defined with respect to the upper structure so it does not move. As shown in Figure 14, the seat is fixed to the upper structure through the revolute joint $S_{10}$, the user can manually adjust the seat inclination, defining an angle $\theta_S$ (that goes up to 2.5°) about the revolute joint $S_{10}$. To aid the user during this process, a removable element represents the position of the point $K^*$. At this stage of the study, the mechanism is not completely designed.

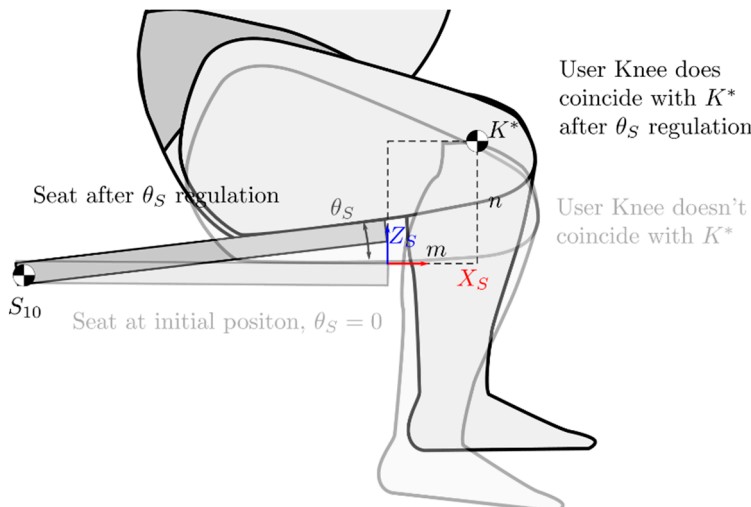

**Figure 14.** Seat inclination mechanism working principle.

## 3. Results

This section summarizes the main results of the design and synthesis processes presented before in terms of parameters, performances, and preliminary design of the prototype of the rehabilitation machine focusing on the leg exercises mechanism and its control system.

### 3.1. Leg Exercises Mechanism

The design of the leg exercise mechanism has followed an iterative method to evaluate the sensitivity to the machine parameters and to size its parts. Among all the parameters introduced before, the leg mechanism is sensitive only to the $TB_0$ distance $r$. Figures 15 and 16 represent the effect of $r$ on the stroke and the force of the pneumatic actuator during the exercise. By increasing the length $r$, it is possible to reduce the required force the actuator has to generate in order to balance the user effort (245 $N$ in the example) and to obtain a more constant behavior, however, this comes at a cost of a longer actuator stroke. Lower forces should be preferable, but the limited available space constrains the design to shorter actuator strokes. The final mechanism parameters are listed in Table 3; they produce the results shown in red in the figures.

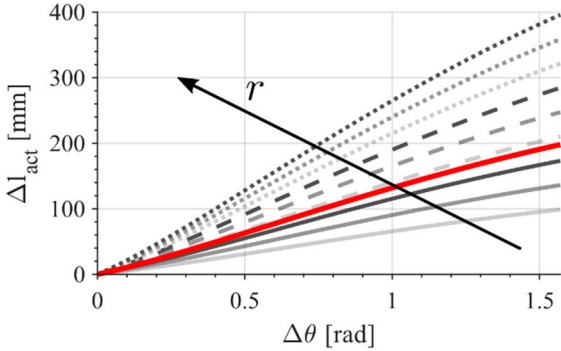

**Figure 15.** Leg exercises mechanism input-output relation sensitivity to *r*.

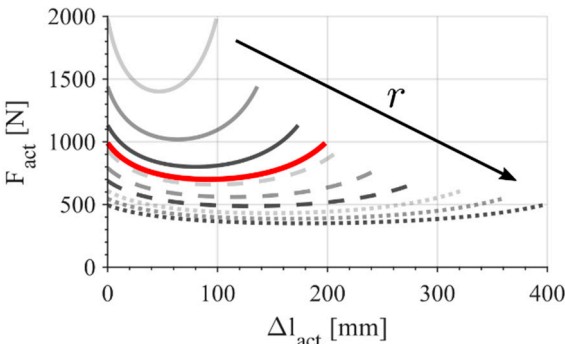

**Figure 16.** Effect of *r* on $F_{act}$ and $\Delta l_{act}$. In red the curve with the final parameters.

**Table 3.** Value of the parameters of the leg exercises mechanism.

| Parameter | Value | Parameter | Value |
|-----------|-------|-----------|-------|
| $a$ | 170 mm | $l_R$ | 400 mm |
| $b$ | 200 mm | $m$ | 91 mm |
| $d$ | 145 mm | $n$ | 107 mm |
| $-h$ | 40 mm | $r$ | 140 mm |
| $-k$ | 90 mm | $\theta$ | 13°–103° |
| $l_{act}$ | 390.0 mm–585.5 mm | $\rho$ | 12.5° |

Figure 17 displays the input-output relation of the mechanism and its instantaneous transmission ratio if the previously listed parameters are used. The actuator stroke $\Delta l_{act}$ as a function of the exercise angle $\Delta \theta$ appears almost linear, as required, but the instantaneous transmission ratio highlights the non-linearity in the relation. Given the linearity requirement and the system constraints, the presented results are satisfactory. The same curves hold for both the leg extension (read the figure from left to right) and the leg curl exercises (from right to left).

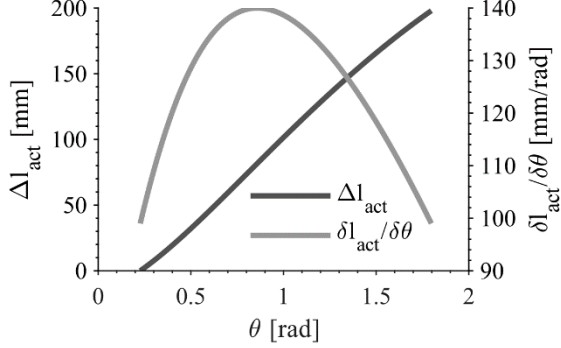

**Figure 17.** Leg mechanism kinematic relation and instantaneous transmission ratio.

Figure 18 depicts the comparison of the force the actuator has to generate in order to balance only the user force (Equation (7)) or the user force, the mechanism weight, and the elastic element (Equation (8)). By considering the weight of the mechanism and the effect of an elastic element, the actuator force is no more symmetric and the difference between leg curl and leg extension exercises becomes clear. In the leg extension exercise, the actuator force is lower than the theoretical one. This can be explained considering that the mechanism weight and the elastic element oppose, together with the actuator force, the user effort to lift the leg. This contribution increases as the angle of the exercise increases; hence the actuator force decreases as the angle increases.

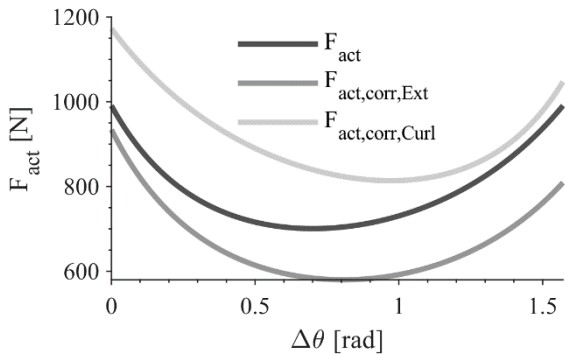

**Figure 18.** Actuator force during the leg motion required to balance a load of 245 N. Comparison between Equations (7) and (8).

On the other hand, the mechanism weight and the elastic element help the user during the curl exercise; therefore, the actuator must generate a higher force to balance the forces acting on the mechanism. The behavior is opposite from before since the weight and the elastic element contributions are more relevant at the beginning of the exercise.

## 3.2. Leg Exercise Control System

In a previous section, the pneumatic circuit designed to generate the required pressure to balance a constant user effort has been described. This pneumatic system has been simulated in order to evaluate the system behavior during the operation. In this model, a pneumatic cylinder with a bore diameter of 40 mm and a rod diameter of 16 mm has been used. The air tank has a volume of 0.024 m$^3$. A simple PID controller has been implemented to drive the proportional flow control valve to regulate the pressure within the cylinder. The PID controller has been chosen due to its simplicity, but if it will prove ineffective or not safe enough during future tests, a more suitable control architecture will be proposed and implemented.

Figure 19 illustrates the result of a leg extension simulation with a desired load of 245 N. The initial pneumatic circuit pressure is 0.15 bar less than the minimum required pressure. About 0.3 rad are needed to reach the set pressure; this value depends on the pneumatic cylinder size since the only way to increase the cylinder pressure is by the user action on the mechanism. The faster the leg moves and the smaller the cylinder is, the quicker the pressure can rise. Each time the cylinder pressure must decrease, the controller opens the proportional valve decreasing the cylinder pressure but increasing the air tank pressure. If the air tank pressure rises above the required pressure, the system stops working correctly. Although a discharge valve is used to avoid such occurrence, each time the valve is operated, the air volume is reduced, so the pressure in the air tank increases even faster. Since the air tank cannot be infinitely large, a correct design of the air tank, the pneumatic cylinder, the proportional valve, and the control system is required to increase the efficiency of the system. Analogous considerations hold for the leg curl exercise.

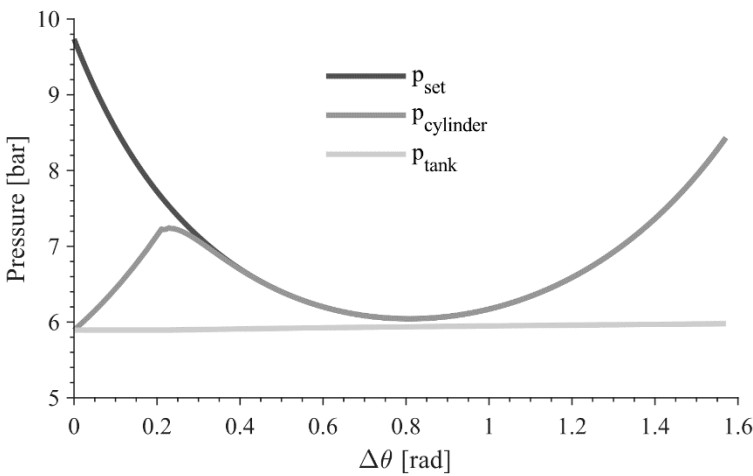

**Figure 19.** Pressure during a leg extension exercise with a load of 245 N.

Control architecture has to be further investigated and revised in order to obtain an optimal control that can satisfy all the requirements while minimizing the pneumatic air waste. For this reason, a PID may not be the most suitable control law.

## 4. Conclusions

This paper describes the initial design phase of an innovative exercise machine for bariatric users. This machine enables obese individuals to increase their energy expenditure with light and simple physical exercises. The novelty is the key requirement of the design: to overcome the user psychological barriers toward typical gym machines, the rehabilitation machine does not appear as such, but it looks like an armchair with all the mechanisms and moving parts hidden.

With the guidance of a medical team from Università degli studi di Torino medical science department, the most suitable exercises have been identified, and the machine mechanisms have been designed to correctly perform these exercises with the requirement that such mechanisms are hidden in order to partially address the issue of obese individuals stepping away from physical exercises. Auxiliary mechanisms have been designed in order to accommodate individuals within a wide range of sizes. The leg exercise mechanism, the upper body exercise mechanism, the structure, and the regulation mechanism are the main sub-systems of the machine, and their functional design has been presented in this study. The leg curl/extension mechanism is a double parallelogram with a virtual joint in correspondence of the user knee that is completely hidden under the seat. To provide the desired load, an interesting pneumatic system is proposed in order to implement an efficient and quiet system. The upper body exercise sub-system is cable-based to guarantee better user mobility and to allow a wide range of exercises. The results analysis focuses mainly on the leg mechanism and its actuation and control system.

The result of this study is an interesting machine that has to be optimized, improved, and finalized in order to then begin an experimental campaign on a prototype. The first prototype will undergo a series of tests to ensure if it is safe to perform clinical trials and if all requirements are met. User safety is our first concern. An improvement and optimization phase will be done while finalizing the prototype design and after the initial sub-systems test stages. Future developments also include an improved control system in order to increase the system efficiency and to reduce the waste of air, the finalization of the upper body exercise sub-system together with its control architecture, and the design of a user interface able to monitor and regulate the user exercise details and to implement some sort of "gamification" in order to motivate the user to exercise even more.

**Author Contributions:** Conceptualization: G.Q.; methodology: A.B., P.C., L.C., C.V. and G.Q.; formal analysis: A.B., P.C., L.C., C.V. and G.Q.; data curation: A.B.; writing—original draft preparation: A.B.; writing—review

and editing: A.B., P.C., L.C., C.V. and G.Q.; visualization: A.B.; supervision: G.Q.; project administration: G.Q.; funding acquisition: G.Q. All authors have read and agreed to the published version of the manuscript.

**Funding:** This research received no external funding.

**Conflicts of Interest:** The authors declare no conflict of interest.

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
