# Peer review of "Rehabilitation Machine for Bariatric Individuals"

_machines, doi:10.3390/machines8030045_

Round 1
Reviewer 1 Report
This paper presented the design, developpement and control of a rehabilitation machine for bariatric individuals. The main contribution of the authors is the developpment of the prototype device focusing on the leg and Upper body exercises mechanism and its control system based on PID law.
The issue investigated is important in health applications. In general, the paper is not well organized about writing and notation. Some parts are difficult to understand since it summarizes different parts on modeling or design without clear methodology.
All rehabilitation devices are controlled by classical control laws. Although the results obtained with these controllers are satisfying in terms of rehabilitation
specifications, they are restrictive in terms of control performance, mainly because they do not theoretically guarantee good behaviour in the entire state space and do not ensure the users safety.
My decision is thus for "Revise and re-submit".
Reviewer 2 Report
Dear Authors,
Well done for writing this interesting paper. Kindly find the following comments to improve your paper:
Abstract should be rewritten to reflect the purpose of the paper and the new approach presented. Current Abstract looks like a part of the Introduction Section.
Include more recent academic studies and update references since many references used are more than 10 years old.
Can you please highlight how this system could improve over existing systems.
Can you please conclude what are the anticipated outcomes from using this system.
To meet safety concerns, I suggest more improvement and optimisation to take place before clinical trials.
Round 2
Reviewer 2 Report
Dear authors,
Well done for improving your paper. I will recommend it to the editorial board.